



# Quasi-separatrix Layers Induced by Ballooning Instability in Near-Earth Magnetotail

Ping Zhu[1,2,3], Zechen Wang[1], Jun Chen[4], Xingting Yan[1], and Rui Liu[4,5]

[1]CAS Key Laboratory of Geospace Environment, Department of Engineering and Applied Physics, University of Science and Technology of China, Hefei, Anhui, China
[2]KTX Laboratory, Department of Engineering and Applied Physics, University of Science and Technology of China, Hefei, Anhui, China
[3]Department of Engineering Physics, University of Wisconsin-Madison, Madison, Wisconsin, USA
[4]CAS Key Laboratory of Geospace Environment, Department of Geophysics and Planetary Sciences, University of Science and Technology of China, Hefei, Anhui, China
[5]CAS Center for Excellence in Comparative Planetology, Hefei, Anhui, China

**Correspondence:** Ping Zhu (pzhu@ustc.edu.cn)

**Abstract.** Magnetic reconnection processes in the near-Earth magnetotail can be highly 3-dimensional (3D) in geometry and dynamics, even though the magnetotail configuration itself is nearly two dimensional due to the symmetry in the dusk-dawn direction. Such reconnection processes can be induced by the 3D dynamics of nonlinear ballooning instability. In this work, we explore the global 3D geometry of the reconnection process induced by ballooning instability in the near-Earth magne-
totail by examining the distribution of quasi-separatrix layers associated with plasmoid formation in the entire 3D domain of magnetotail configuration, using an algorithm previously developed in context of solar physics. The 3D distribution of quasi-separatrix layers (QSLs) as well as their evolution directly follows the plasmoid formation during the nonlinear development of ballooning instability in both time and space. Such a close correlation demonstrates a strong coupling between the ballooning and the corresponding reconnection processes. It further confirms the intrinsic 3D nature of the ballooning-induced plasmoid
formation and reconnection processes, in both geometry and dynamics. In addition, the reconstruction of the 3D QSL geometry may provide an alternative means for identifying the location and timing of 3D reconnection sites in magnetotail from both numerical simulations and satellite observations.

## 1 Introduction

There has been a long standing controversy over whether the magnetic reconnection or the ballooning instability in the magne-
totail actually triggers the onset of substorms, since both mechanisms found support in observation and simulation(e.g. Baker et al. (1996); Lui (1991); Angelopoulos et al. (2008); Panov et al. (2012)). To resolve the controversy, it may be necessary to study and understand the evolution of the magnetotail in the substorm growth phase, and to identify and predict the signatures of magnetic reconnection and ballooning instability in the magnetotail, as well as their potential connections. In practice, the conventional two-dimensional reconnection models with spatial symmetries in both in-flow and out-flow regions are often used
to identify and interpret the signatures of reconnection process from observational data. However, one fundamental question





that remains to be addressed is whether the magnetic reconnection in magnetotail, when it does occur, can be always interpreted in the conventional two-dimensional picture, and if not, how one may characterize its intrinsically three-dimensional geometry.

The overall evolution of the magnetotail-like configuration has been studied for many years ( (Schindler, 2007) and references therein). In particular, the plasmoid formation process was investigated in details in earlier 3D resistive MHD simulations
by Birn and Hones, Jr. (1981) and by Hesse and Birn (1991), for example. Recently, our simulations based on the 3D full MHD equations implemented in the NIMROD code (Sovinec et al., 2004) have found a plasmoid formation process in the generalized Harris sheet that is often used as an approximate configuration of the near-Earth magnetotail prior to a substorm onset (Zhu and Raeder, 2013, 2014). Those simulations demonstrate that the embedded thin current is unstable to ballooning mode perturbations, and the nonlinear development of the ballooning instability is able to induce the onset of reconnection and the formation
of plasmoids in the current sheet where there is no pre-existing X-point or X-line.

In comparison to the low-$S$ (i.e. Lundquist number) regime, where $S \sim 10^2$ considered in the earlier simulations by Birn and Hones, Jr. (1981), our recent simulations are in higher-$S$ regime, where $S \geq 10^4$, which may be more relevant to the collisionless regime of plasmas in the magnetotail. In the low-$S$ regime, the magnetotail plasma is linearly unstable to resistive tearing modes, and the associated reconnection process is initially a linear process. In contrast, in the higher-$S$ regime considered in
our recent work (Zhu and Raeder, 2013, 2014), the generalized Harris sheet is linearly stable to resistive tearing modes. The onset of reconnection is a consequence of the nonlinear development of ballooning instability, and the subsequent reconnection is a nonlinear process. Thus, the reconnection processes in the low-$S$ regime reported in the earlier work by Birn and Hones, Jr. (1981) and by Hesse and Birn (1991) are essentially 2D, whereas the reconnection process in the higher-$S$ regime in our simulations is an intrinsically 3D process that does not exist in the 2D geometry. This key difference distinguishes our recent
work (Zhu and Raeder, 2013, 2014) from the previous work by Birn and Hones, Jr. (1981) and by Hesse and Birn (1991).

Although our previous work has demonstrated in MHD simulations the formation of plasmoids induced by ballooning instability in the generalized Harris sheet (Zhu and Raeder, 2013, 2014), the global 3D structure of the ballooning induced reconnection was not clear. In particular, the reconnection process in our simulations is no longer invariant along the equilibrium current direction, unlike in a conventional 2D reconnection process. This leads to general questions as to where and how
reconnection takes place in the 3D configuration, as well as how the global structure of the 3D reconnection process can be characterized and captured in manners different from the more familiar 2D reconnection process. More fundamentally, it has remained unclear whether this 3D reconnection process can be reducible to or interpretable in terms of the conventional 2D reconnection processes.

Whereas the overall evolution of the magnetotail-like configuration has been studied in space community for many years,
the irreducible dimensionality of the reconnection process associated with the evolution of ballooning instability has never been addressed before in literature, including the papers by, e.g. Birn and Hones, Jr. (1981); Hesse and Birn (1991) which were reviewed in the book by Schindler (2007). There is also a long history of work trying to identify the possible role of out-of-plane instabilities on reconnection (see for example, Pritchett (2013) and Sitnov et al. (2014)). Different from those previous work, in this work we intend to identify the geometry features associated with the intrinsically 3D reconnection process induced by
the ballooning instability in near-Earth magnetotail, in light of those questions raised in the previous paragraph.



The quasi-separatrix layer (QSL) is a concept we adopt for the above purposes. In fact, QSL has long been a common and powerful tool for the analysis and understanding of magnetic structures in the solar atmosphere (Titov and Démoulin, 1999; Titov et al., 2002). Recently the concept of QSL has also been effectively applied to the analysis of laboratory reconnection experiments (Lawrence and Gekelman, 2009). Previously, we calculated the spatial distribution and the structure of the QSLs, as well as their temporal emergence and evolution, within the equatorial plane (Zhu et al., 2017), based on the earlier simulation results on the formation of plasmoids induced by ballooning instability in the magneotail (Zhu and Raeder, 2013, 2014). There we found the QSL structures are not invariant along any direction within the 2D equatorial plane; instead they are disconnected and isolated local structures. Those initial findings start to reveal the intrinsic 3D nature of the reconnection induced by ballooning instability in the generalized Harris sheet, which is irreducible to 2D reconnection process in geometry and dynamics within the 2D equatorial plane. In this work, we extend our previous study within the 2D equatorial plane to the entire 3D domain of near-Earth magnetotail. Using a newly developed implementation for efficiently computing the squashing degree of magnetic field lines in any 3D domain (Liu et al., 2016), we obtain the 3D distribution of QSLs as well as their evolution in the near-tail plasma sheet. The intersection of the 3D distribution of QSLs with equatorial plane recovers results from our previous work. More importantly, the calculated 3D distribution of QSLs provides a complete and global view of the geometric structure of the 3D reconnections associated with the plasmoid formation induced by the nonlinear ballooning instability in the near-Earth magnetotail.

The rest of the paper is organized as follows. We first briefly review our previous simulation results for the plasmoid formation process induced by ballooning instability in Sec. 2. Next in Sec. 3 we describe the method we use for efficiently evaluating the squashing degrees of entire magnetic fields. Both 2D and 3D distributions of QSLs revealed from the squashing degree calculation are reported and analyzed in Sec. 4. Finally, summary and discussion are given in Sec. 5.

## 2 Plasmoid formation induced by ballooning instability

Our recent MHD simulations are developed to demonstrate the dynamic process of plasmoid formation induced by nonlinear ballooning instability of the near-Earth magnetotail. In these simulations, the magnetic configuration of near-Earth magnetotail is modeled using the generalized Harris sheet, which can be defined in a Cartesian coordinate system as $\mathbf{B}_0(x,z) = \mathbf{e}_y \times \nabla\Psi(x,z)$, $\Psi(x,z) = -\lambda \ln \dfrac{\cosh\left[F(x)\frac{z}{\lambda}\right]}{F(x)}$, $\ln F(x) = -\int B_{0z}(x,0)dx/\lambda$, and $\lambda$ is the characteristic width of the current sheet. The conventional Harris sheet is recovered when $F(x) = 1$. The configuration can be further specified with a particular $B_z$ profile that features a minimum region along the $x$ axis, corresponding to an embedded thin current sheet (Fig. 1), such as those often found in global MHD simulations and inferred from satellite observations in the near-Earth magnetotail.

For a sufficiently small magnitude of $B_z$ minimum, the magnetotail becomes unstable to ballooning instability, whose nonlinear development leads to the formation of tailward receding plasmoids in the magnetotail (Fig. 2). The magnetic reconnection process in these simulations is no longer invariant along the equilibrium current direction, unlike in a conventional 2D reconnection process. For example, at a time after the formation of plasmoids, those field lines crossing the $y = -90$ line in the $z = 0$ plane encounters totally different plasmoid structure from the field lines crossing the $y = -95$ line in the $z = 0$ plane



(Fig. 3). Questions arise as to where and how a reconnection takes place in the 3D configuration, as well as how the global structure of the 3D reconnection process can be characterized and captured in manners different from the more familiar 2D reconnection process. Further, it remains unclear whether this 3D reconnection process can be reducible to or interpretable in terms of the conventional 2D reconnection processes.

## 3   Methodology

To address these questions in this work, we for the first time, apply the concept of quasi-separatrix layer (QSL) to the analysis of the geometry of magnetic reconnection induced by ballooning instability in a generalized Harris sheet that represents the magneotail. QSL has been adopted for the analysis of the reconnection structures involved in the solar corona for a long time (e.g. Titov and Démoulin (1999); Titov et al. (2002)). It has also been effectively applied to the analysis of laboratory reconnection experiments (Lawrence and Gekelman, 2009). A QSL is a 3D structure defined by steep gradient in the field line connectivity, which are quantified by mapping field lines across a specified volume. A surface, $S$, must first be defined to enclose this volume. Divide $S$ into two subspaces, $S_0$ and $S_1$, where $S_0$ and $S_1$ represent the surfaces on which field lines enter and leave the volume respectively. The initial footpoint is defined as $\mathbf{r}_0 = (u_0, v_0)$ in $S_0$. One then traces the field line from the initial footpoint through the enclosed volume until the field line leaves the volume through $S_1$ at the point $\mathbf{r}_1 = (u_1, v_1)$. The Jacobian transformation matrix and the norm of the mapping from $(u_0, v_0)$ to $(u_1, v_1)$ are defined as

$$\mathcal{J} = \frac{\partial \mathbf{r}_1}{\partial \mathbf{r}_0} = \begin{pmatrix} \frac{\partial u_1}{\partial u_0} & \frac{\partial u_1}{\partial v_0} \\ \frac{\partial v_1}{\partial u_0} & \frac{\partial v_1}{\partial v_0} \end{pmatrix} \tag{1}$$

$$N = \sqrt{\left(\frac{\partial u_1}{\partial u_0}\right)^2 + \left(\frac{\partial u_1}{\partial v_0}\right)^2 + \left(\frac{\partial v_1}{\partial u_0}\right)^2 + \left(\frac{\partial v_1}{\partial v_0}\right)^2}. \tag{2}$$

A QSL is the region where the gradient of this mapping is large compared to the average mapping, i.e. $N >> 1$.

Mathematically, the squashing degree $Q$ is defined as $Q = N^2/|\Delta|$ where $\Delta$ is the determinant of the Jacobian matrix (Titov et al., 2002; Priest and Demoulin, 1995). The variation of $Q$ among different field lines reflects the deformation of the magnetic flux tubes. A high squashing degree corresponds to a large variation in the cross-sectional area of an elemental flux tube from one footpoint to another. Quasi-separatrix layers turn into separatrices in the limit the layer thickness goes to zero, or the corresponding squashing degree goes to infinity. The physical significance of QSL is that current sheets preferentially form on these layers for reconnection.

A newly developed implementation for efficiently computing the squashing degree of magnetic field lines in any 3D domain has been successfully applied to investigating the evolution of magnetic flux ropes in coronal magnetic field extrapolated from photospheric magnetic field (Liu et al., 2016). The method utilizes the field-line mappings between a cutting plane and the footpoint planes to give optimal results for mapping the squashing factor in the cutting plane. In order to avoid spurious high squashing degree structures for field lines touching the cutting plane, a new plane perpendicular to the particular field line can be introduced and switched to using the same method. We adopt this new method to recover our previous results on 2D QSL




distribution based on the calculation of bald patches. We further use the new method to find the 3D distribution of QSLs in the entire domain.

## 4 Major results

In this section, we compute the squashing degrees and analyze the 2D and 3D QSL distribution of the magnetic field config-
uration as well as it evolution in the near-Earth magnetotail, in an attempt to understand the global geometry of the magnetic field and the 3D nature of the magnetic reconnection process in association with the plasmoid formation process induced by ballooning instability.

### 4.1 2D spatial distribution of QSLs in equatorial plane

We first review the development of QSLs in the equatorial plane of magnetotail (i.e. $z = 0$ plane) based on the computation
of squashing degrees, as shown in Fig. 4, for the same time sequence of nonlinear ballooning development that leads to the formation of tailward receding plasmoids in the magnetotail (Fig. 2). Similar results on QSLs are also obtained in our previous work, where the QSLs are identified based on the computation of bald patches (Zhu et al., 2017). Here the QSLs are identified as the boundaries of white patches in a plane, on which the squashing degree becomes singularly large.

In the initial and early stage of ballooning instability evolution, QSLs are absent in the $z = 0$ plane ($t = 170$) (Fig. 4, upper
left). By the time $t = 180$ the first set of QSLs denoted as the white enclosed regions start to form periodically along the $y$ direction within the $z = 0$ plane around the line of $x = 9.5$ (Fig. 4, upper right). As the ballooning instability continues to evolve, a second set of QSLs start to form in the equatorial plane near the radially extending fronts of ballooning fingers around $x \lesssim 13.5$ ($t = 190$) (Fig. 4, middle left). The circular shape of each of these QSLs is smaller in radius than the first set of QSLs. Their spatial distribution pattern is similar to the first set of QSLs, but their locations are shifted in $y$ direction from the first
set by one half distance between two adjacent QSLs. After reaching their maximum sizes, the first set of QSLs begin to shrink into ellipses squeezed in the $x$ direction and eventually disappear ($t = 220 - 260$) (Fig. 4, middle right, lower left, and lower right). In addition, the locations of the QSLs also evolve, particularly those of the second set. As the ballooning finger tips extend in the positive $x$ direction, the QSLs behind the each finger tip in the second set move along in the same tail direction. Furthermore, as the first set of QSLs nearly shrink into disappearance, a third set of QSLs start to emerge at $x = 11$ between
the first two sets around $t = 240$ (Fig. 4, lower left). This set of QSLs later become dominant in size after the first set disappear and the second set also shrink in size. Different from the first set, the third set of QSL circles have the same locations in $y$ direction as those in the second set. The timings and locations of the emergence of these QSL structures, correlate well with those of the plasmoid development as shown in Fig. 2.

Even within the 2D equatorial plane ($z = 0$), the isolated and discrete distribution of QSLs in both $x$ and $y$ directions indicates
the 3D feature of the corresponding reconnection process. In another word, the X-line in conventional 2D reconnection has broken into a group of disconnected locations of reconnections as represented by QSLs. A close examination of one of the QSLs centered around $x = 9.5, y = 15$ and another centered around $x = 13.5, y = 10$ at $t = 190$, finds that the variation of



squashing degree at the QSL on the boundary of an isolated region is rather spiky instead of smooth (Fig. 5). Away from the QSL, the logarithms of squashing degree are close to zero and their variation is flat and smooth. The QSL structures are indeed located surrounding well isolated regions, which are outcome of the irreducible 3D nature of the corresponding reconnection process.

## 4.2  3D spatial distribution of QSLs

We further examine the 3D distribution of QSLs in the entire simulation domain of magnetotail. Not only are QSLs located in isolated regions in 2D plane, they are also localized in isolated and confined regions in 3D domain (Fig. 6). As shown in Fig. 6, the circles representing QSLs in 2D plane are extended to the iso-surfaces representing QSLs in 3D space. Such regions of QSLs are localized along the equilibrium field line near the equatorial plane, such as those shown in Fig. 1 (lower panel). This is consistent with field line structure during the nonlinear development of ballooning instability, where the plasmoids are centered around the equatorial plane with north-south ($z$) symmetry. The distribution of the QSL structures are periodic along the west-east ($y$) direction (Figs. 7 and 8), same as the QSL distribution within the 2D equatorial plane. The 3D distribution of QSLs provides a global and complete view of where the reconnection takes place. They further confirm the irreducible 3D nature of the corresponding reconnection process.

Another approach to characterizing the 3D distribution of QSLs in the near-Earth magnetotail, is to examine the squashing degree contours on various strategically selected 2D slices parallel or perpendicular to coordinate axes. For example, at an earlier time $t = 190$, the squashing degree distributions in the $y - z$ planes show two elliptically shaped QSL regions centered around $(y, z) = (5, 0)$ at $x = 9.45$ and $(y, z) = (10, 0)$ at $x = 13.38$ respectively, which again are represented by the white space where the squashing degree becomes singular (Fig. 9, upper row). In the $x - z$ plane, the corresponding two QSL regions manifest themselves as two round areas of singular squashing degree located around $(x, z) = (9.45, 0)$ at $y = 5$ and $(x, z) = (13.38, 0)$ at $y = 10$ (Fig. 9, middle row). In the $x - y$ planes with equal distance off the equatorial plane ($z = -0.03$ and $z = 0.03$), the QSL regions are similar to those within the equatorial plane shown in Fig. 4 in both location and shape, and the QSL distribution in those two $x - y$ plane are symmetric with respect to $z = 0$ (Fig. 9, lower row). However, those QSL regions disappear as the $x - y$ planes move further away from the equatorial plane, indicating the localized nature of the 3D reconnection regions.

25

The above approach also helps visualizing the development of 3D distribution of QSLs over time. At a later time $t = 240$, three QSL regions appear along $x$ axis at $x = 9.25$, $11.0$, and $14.3$, which can be first seen from the squashing degree contours within the $y - z$ planes (Fig. 10, upper row). This is in contrast with the earlier time at $t = 190$, when QSLs only appear in two $y - z$ planes along the $x$ axis (Fig. 9, upper row). At the same time, the three QSL regions also show up in the $x - z$ planes, individually or together, depending on where the plane locates in the $y$ direction (Fig. 10, middle row). For example, the two QSL regions in $x - z$ plane around $(x, z) = (11.0, 0)$ and $(x, z) = (14.3, 0)$ (Fig. 10, middle row, right panel) correspond to the two QSL regions in $y - z$ plane around $(y, z) = (10.0, 0)$, but one in $x = 11.0$ plane (Fig. 10, upper row, middle panel) and another in $x = 14.3$ plane (Fig. 10, upper row, right panel) respectively. Furthermore, the time development of QSL 3D distribution can be also viewed from the variation of squashing degree contours in the $x - y$ planes along $z$ direction (Fig. 10,

30



lower row). In particular, in comparison to the earlier time at $t = 190$, the dominant QSL regions have shifted from around $(x, y) = (9, 5)$ (Fig. 9, lower row) to about $(x, y) = (14.3, 10)$ near the $z = 0$ equatorial plane by the time $t = 240$ (Fig. 10, lower row). Together, and over time, these slices with different but complementary orientations compose a complete views about the development of the global 3D distribution of QSLs. In comparison with the timings and locations of the emergence

of the plasmoid development shown in Fig. 2, one can see that 3D distribution of QSLs as well as their evolution directly follows the plasmoid formation during the nonlinear development of ballooning instability in both time and space. More importantly, the 3D QSL distribution and evolution provide a more global and complete view of the 3D geometry of magnetic reconnection process induced by the nonlinear ballooning instability in the near-Earth magnetotail.

## 5 Summary and discussion

In summary, the 3D distribution of quasi-separatrix layers (QSL), as well as its evolution directly following the nonlinear development of ballooning instability in the near-Earth magnetotail, has been thoroughly evaluated and examined based on previous resistive MHD simulation data on the plasmoid formation process induced by the ballooning instability. The quasi-separatrix layers have been identified by locating the regions of high squashing degree throughout the entire 3D domain of the model near-Earth magnetotail in simulation. It is found that the 3D distribution of QSLs correlates well not only with

the 2D mode structures of ballooning instability within the $x - y$ plane, but also with the 3D ballooning mode structures as projected onto $x - z$ and $y - z$ planes, both spatially and temporally during the evolution of the magnetotail configuration. Such a close correlation demonstrates a strong coupling between the ballooning and the corresponding reconnection processes. It also further confirms the intrinsic 3D nature of the ballooning-induced plasmoid formation and reconnection processes, in both geometry and dynamics. In addition, the reconstruction of the 3D QSL geometry may provide an alternative means

for identifying the location and timing of 3D reconnection sites in magnetotail from both numerical simulations and satellite observations.

Whereas the near-Earth magnetotail can become ballooning unstable under substorm conditions, the nonlinear evolution of ballooning instabilities, by themselves, may not lead to the near-explosive substorm onset. Previous studies (Pritchett and Coroniti, 1999, 2010, 2013; Zhu et al., 2004), have demonstrated the persistent presence of ballooning instabilities in general-

ized Harris sheet and magnetotail configurations. The models have varied from the global scales in the ideal MHD models, to the meso scales of 2-fluid models, and eventually to the microscopic scales of kinetic models of plasmas. Since the intrinsic 3D nature of the reconnection process reported in this work derives from the nature of ballooning instability, the global 3D geometry structure of the ballooning-induced reconnection process is expected to persist in presence of 2-fluid and kinetic effects, particularly on the macroscopic scales where both MHD and kinetic models should agree.

Although this work was in part motivated by the substorm problem in magnetospheric physics, it should not be seen as one confined only to the space plasma physics community. Rather, with our first application of QSL to the magnetotail configuration represented by the generalized Harris sheet, this work provides new insight into the ubiquitous 3D reconnections in nature and laboratory by identifying and characterizing 3D reconnection induced by ballooning instability.



Because the 2D perception of magnetic reconnection has been the conventional paradigm for interpreting and understanding most phenomena and processes associated with reconnection in both natural and laboratory plasmas since the beginning, our work and results provide a dramatically different and refreshing view on one of the most fundamental processes in all plasmas. It touches the core question as to what exactly defines a reconnection, or whether reconnection in two dimension and

5  three dimension are qualitatively different. Different answers to such a question can lead to vastly contrasting or contradicting interpretations and conclusions. These issues would continue to be addressed in future work.

*Acknowledgements.* This research was supported by National Natural Science Foundation of China grant No. 41474143, the 100 Talent Program of Chinese Academy of Sciences, and the U.S. DOE grant Nos. DE-FG02-86ER53218 and DE-FC02-08ER54975. The computa-tional work used the NSF XSEDE resources provided by TACC under grant number TG-ATM070010, and the resources of NERSC, which

10 is supported by DOE under Contract No. DE-AC02-05CH11231.





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





**Figure 1.** $B_z(x, z = 0)$ profile (upper) and magnetic field lines (lower) of a generalized Harris sheet as a proxy to the near-Earth magnetotail configuration.





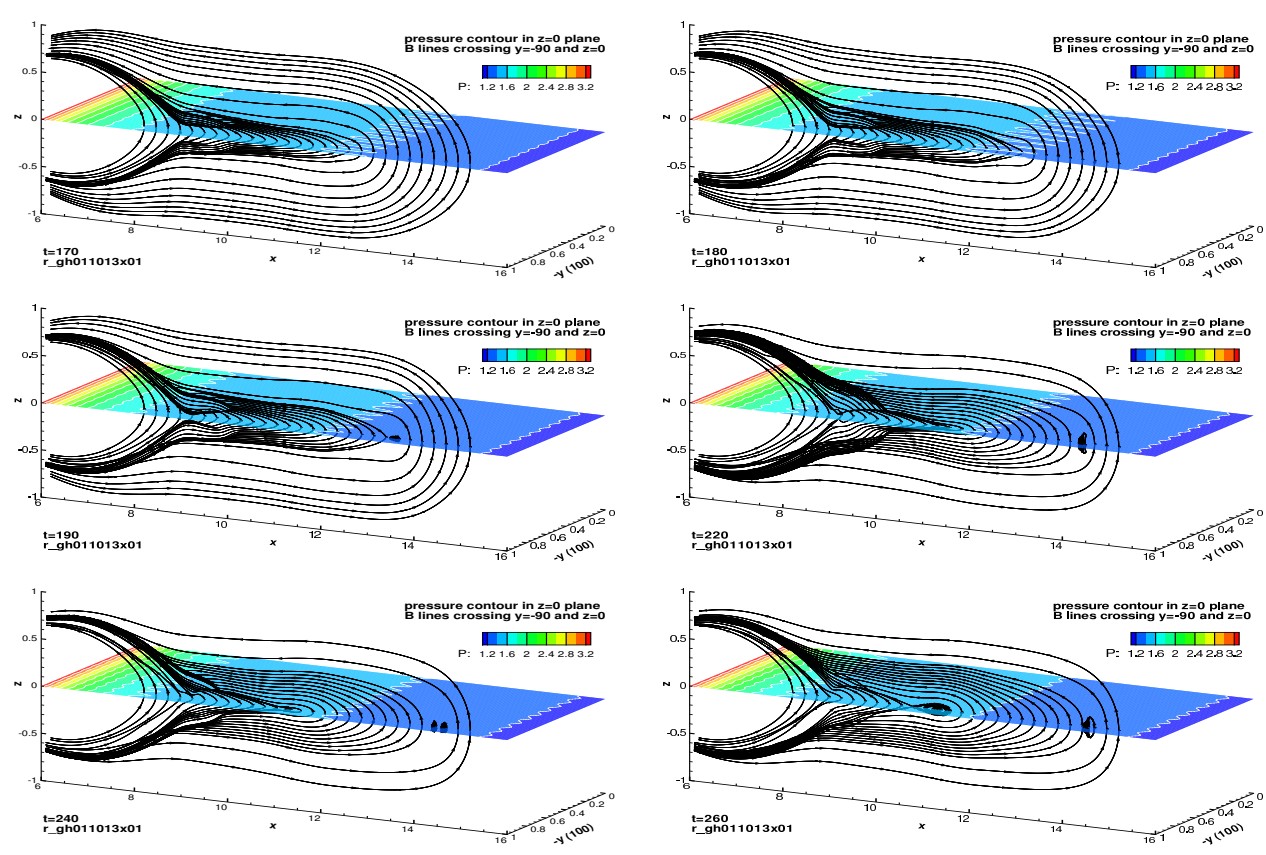

**Figure 2.** Total pressure contours in $z = 0$ plane and magnetic field lines crossing the intersection of $z = 0$ and $y = -90$ planes at selected time slices ($t = 170, 180, 190, 220, 240, 260$). The unit of all coordinate axes is Earth radius $R_{\mathrm{E}}$. The time unit is Alfvénic time $\tau_A$.



**Figure 3.** Magnetic field lines crossing lines $y = -90, z = 0$ (upper) and $y = -95, z = 0$ (lower), and pressure contours in the $z = 0$ plane at $t = 200$. The unit of all coordinate axes is Earth radius $R_{\mathrm{E}}$. The time unit is Alfvénic time $\tau_A$.



**Figure 4.** Contours of the logarithm of squashing degree in $z = 0$ plane at $t = 170$ (upper left), $t = 180$ (upper right), $t = 190$ (middle left), $t = 220$ (middle right), $t = 240$ (lower left), $t = 260$ (lower right). White circles denote the locations where the squashing degree becomes singular.





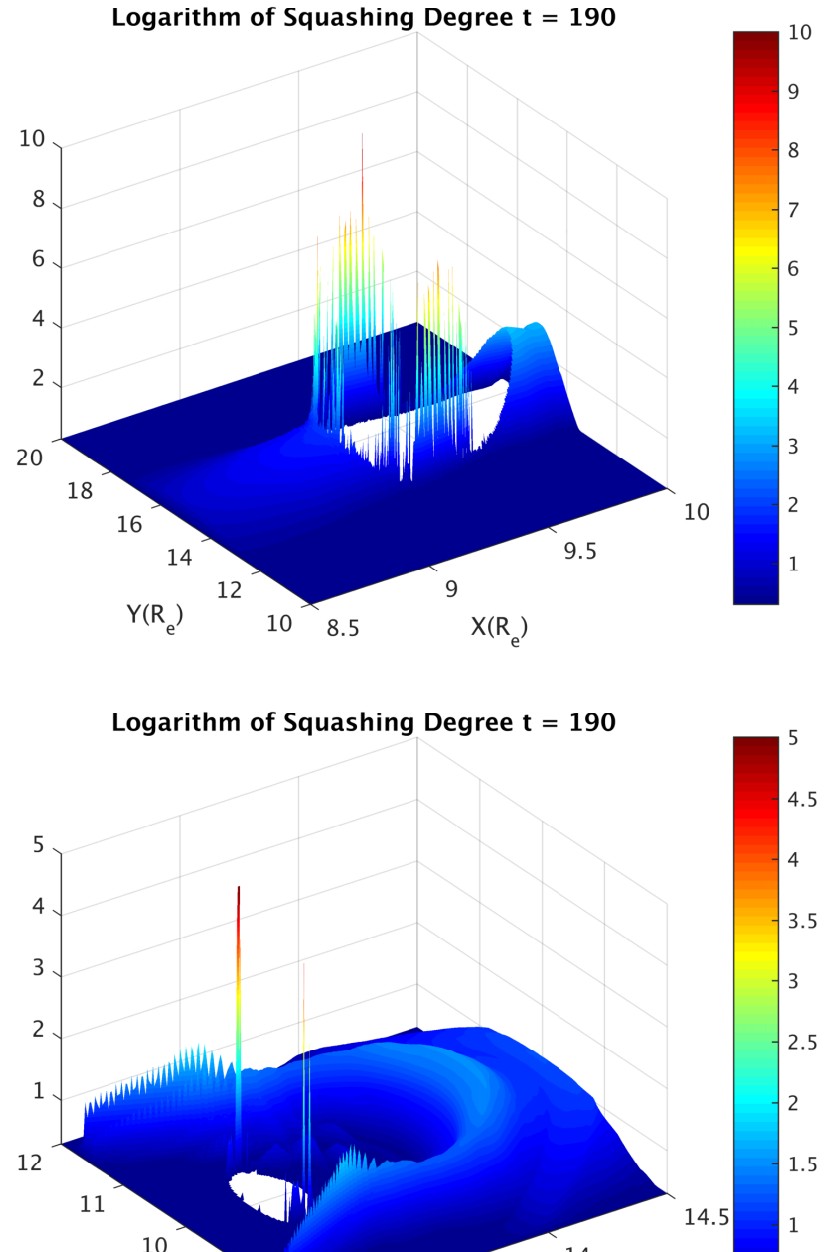

**Figure 5.** Surface plots for the logarithm of squashing degree in the $z = 0$ plane around $x = 9.5, y = 95$ (upper) and $x = 13.4, y = 90$ (lower) at $t = 190$.

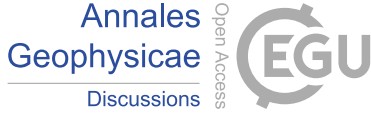



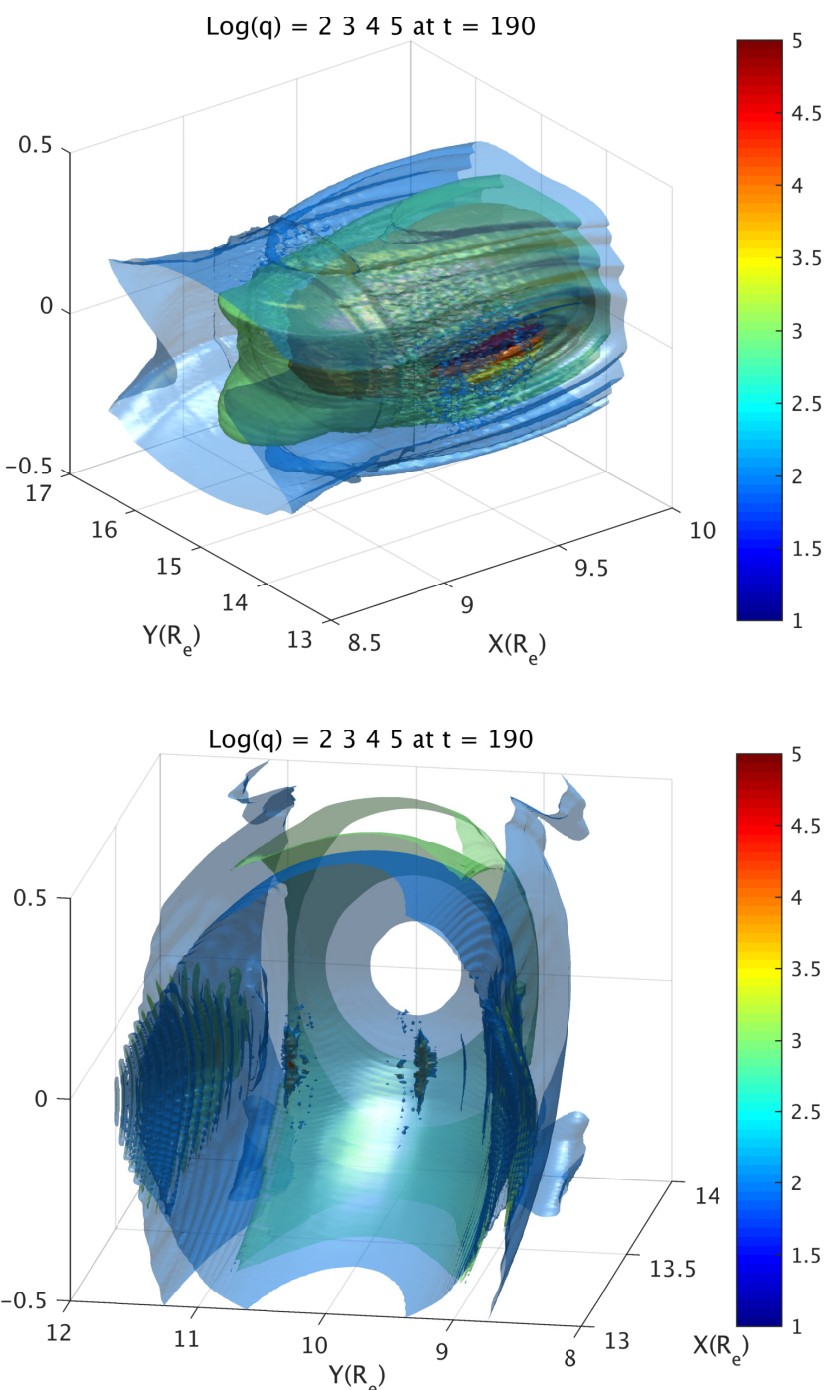

**Figure 6.** Iso-surfaces of the logarithm of squashing degree in the 3D domains centered at $x = 9.5, y = 15, z = 0$ (upper) and $x = 13.4, y = 10, z = 0$ (lower) respectively at $t = 190$.





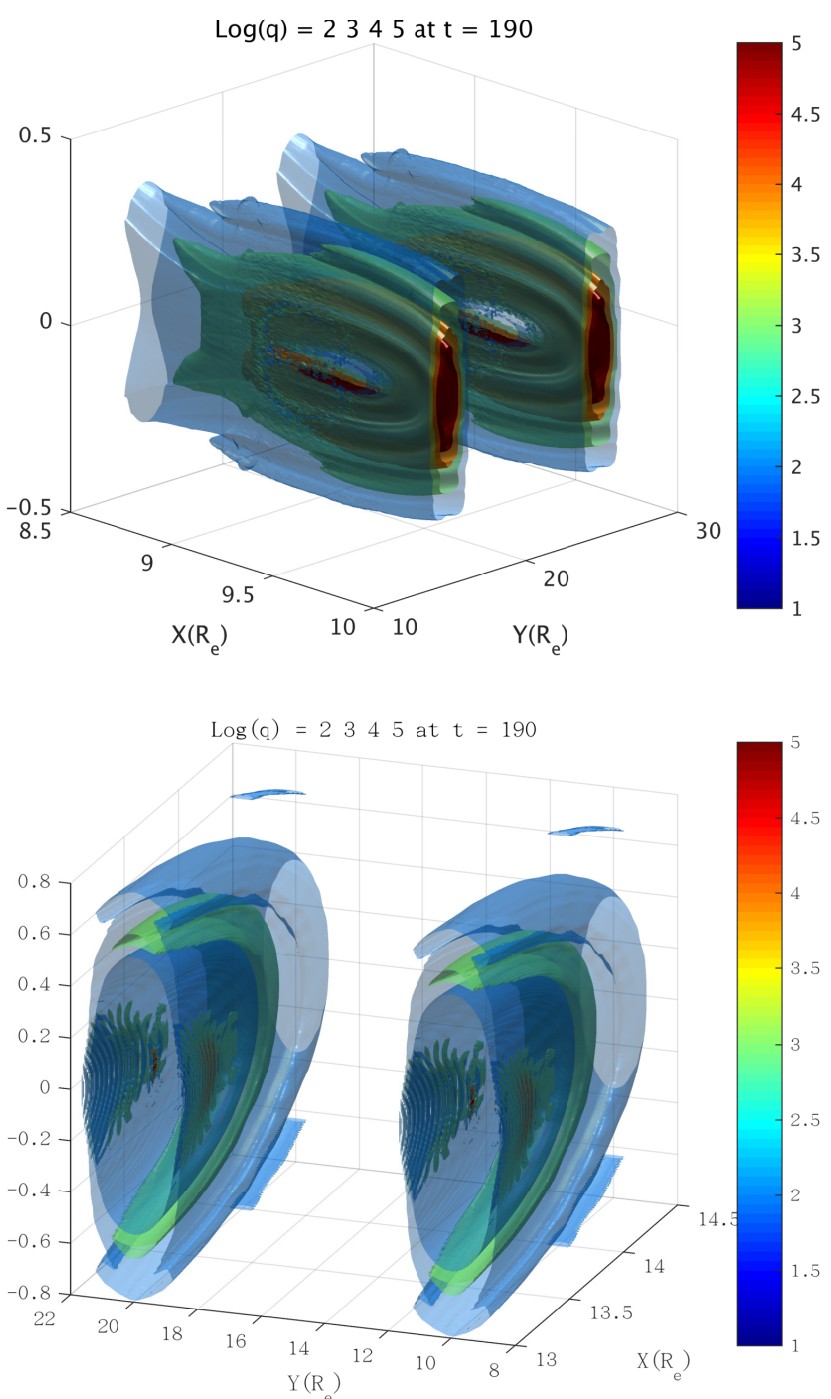

**Figure 7.** Iso-surfaces of the logarithm of squashing degree in the broader 3D domains, which include 2 periods of repeating QSL distribution from $x = 9.5, y = 15, z = 0$ to $x = 9.5, y = 25, z = 0$ (upper) and from $x = 13.4, y = 10, z = 0$ to $x = 13.4, y = 20, z = 0$ (lower) respectively at $t = 190$.

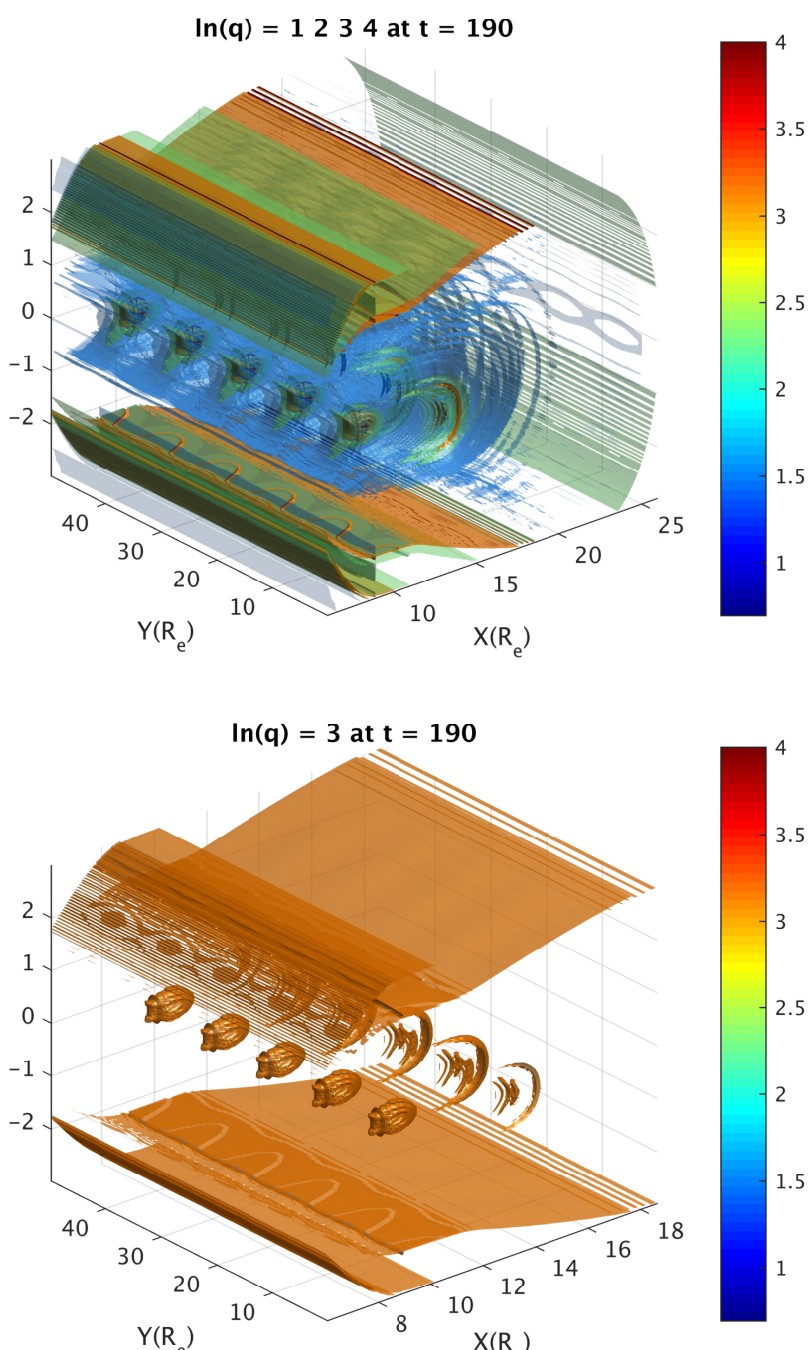

**Figure 8.** Upper: Iso-surface of the logarithm of squashing degree in the broader 3D domains, which include 5 periods of repeating QSL distribution at $t = 190$; Lower: same as upper panel, except that only the iso-surface of the logarithm of squashing degree equaling to 3 is plotted.

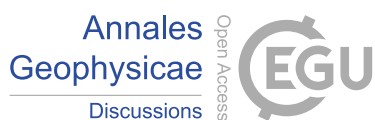



**Figure 9.** Contours of the logarithm of squashing degree in the $x = 9.45$ and $x = 13.38$ planes (upper); in the $y = 5$ and $y = 10$ planes (middle); and in the $z = -0.03$ and $z = 0.03$ planes (lower) at $t = 190$.

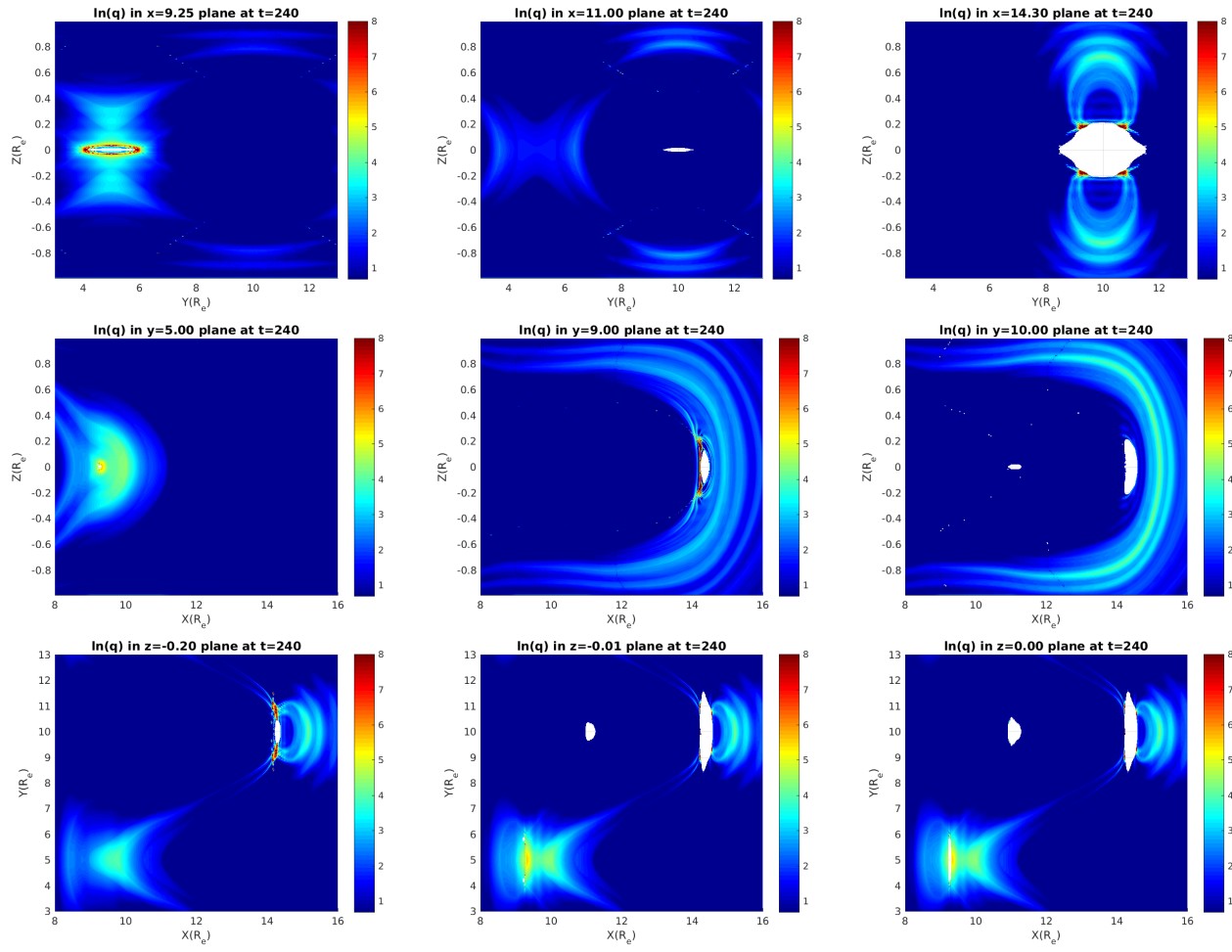

**Figure 10.** Contours of the logarithm of squashing degree in the $x = 9.25$, $x = 11$, and $x = 14.3$ planes (upper); in the $y = 5$, $y = 9$, and $y = 10$ planes (middle); and in the $z = -0.2$, $z = -0.01$, and $z = 0$ planes (lower) at $t = 240$.