# Peer review of "Quasi-separatrix Layers Induced by Ballooning Instability in Near-Earth Magnetotail"

_Annales Geophysicae, 2019_

## Referee Comment (RC1) · Andrei Runov (Referee) · 6 Feb 2019

The research paper by Ping Zhu et al. presents results of probing 3-D geometry of magnetotail reconnection by examining distribution and evolution of quasi-separatrix layers (QSL) in the MHD simulation output. The QSL analysis technique was developed to study magnetic structures in the solar corona. This method has also been used to study 3-D topology of the magnetic field in laboratory plasma experiments. According to my knowledge, it is the first attempt to apply the QSL analysis to simulated data of the magnetotail. The paper shows that the method, indeed, applicable and may be used for analysis of 3-D magnetic topology. The paper is well written, the presentation of results is clear. On this basis, the paper by Ping Zhu et al. is suitable for publication, basically, in its present form. Yet, a couple of points in Summary and Discussion needs

clarification.

In line 19, page 7 the Authors stated: "... the reconstruction of the 3D QSL geometry may provide an alternative means for identifying the location and timing of 3D reconnection sites in magnetotail from both numerical simulations and satellite observations." It is very interesting statement, and I would ask the Authors to comment possible applications of their methods to in-situ observations. If my understanding of the method is correct, the Jacobian transformation matrix and the norm should be defined within the entire region of space to calculate QSL. Am I correct? In the other words, the knowledge of the magnetic field lines connectivity is required. Obviously, it is not the case for single-point spacecraft measurements. May the requirement be fulfilled in a case of multi-point observations? Would 4-points observations (Cluster, MMS) be sufficient when the probe tetrahedron crosses the region of interest?

Later the Authors state: "Whereas the near-Earth magnetotail can become ballooning unstable under substorm conditions, the nonlinear evolution of ballooning instabilities, by themselves, may not lead to the near-explosive substorm onset." This statement is somewhat out of context. Do the Authors mean that coupling between ballooning and reconnection is necessary for explosive-like process?

---

## Author Comment (AC1) · 17 Feb 2019

We thank the referee Dr. Andrei Runov for the encouraging evaluation and recommendation on our work and manuscript. Below we clarify each of the points raised by the referee:

1."In line 19, page 7 the Authors stated: "... the reconstruction of the 3D QSL geometry may provide an alternative means for identifying the location and timing of 3D recon- nection sites in magnetotail from both numerical simulations and satellite observations." It is very interesting statement, and I would ask the Authors to comment possible applications of their methods to in-situ observations. If my understanding of the method is correct, the Jacobian transformation matrix and the norm should be defined within the entire region of space to calculate QSL. Am I correct? In the other words, the knowledge of the magnetic field lines connectivity is required. Obviously, it is not the case for single-point spacecraft measurements. May the requirement be fulfilled in a case of multi-point observations? Would 4-points observations (Cluster, MMS) be sufficient when the probe tetrahedron crosses the region of interest?"

Reply: Yes, it is correct that the knowledge of the magnetic field lines connectivity is required for the calculation of QSL. The in-situ observation data from both single-point and multi-point spacecraft measurements, with additional assumptions and modeling, have been used in various reconstruction methods for the magnetic field line geometry in magnetotail. These include the global MHD simulations of magnetotail evolution calibrated using the in-situ observation data in general (e.g. [Raeder et al 2008]), the Grad-Shafranov (GS) method for two-dimensional (2D) magnetohydrostatic structure based on single-spacecraft data analysis technique (e.g. [Hasegawa et al 2013]), and the magnetic field rotation analysis (MRA) method based on four-point measurements of the magnetic field (e.g. [Shen et al 2007]). The reconstructed region of interest using these methods and in-situ observation data can then be subject to the calculation of QSL.

Raeder, J., Larson, D., Li, W., Kepko, L., and Fuller-Rowell, T., OpenGGCM Simulations for the THEMIS Mission, Space Sci. Rev., 141, 535■555, doi:10.1007/s11214-008-9421-5, 2008.

Hasegawa, H., B. U. Ö. Sonnerup, Q. Hu, and T. K. M. Nakamura (2014), Reconstruction of an evolving magnetic flux rope in the solar wind: Decomposing spatial and temporal variations from single-spacecraft data, J. Geophys. Res. Space Physics, 119, doi:10.1002/2013JA019180.

Shen, C., X. Li, M. Dunlop, Q. Q. Shi, Z. X. Liu, E. Lucek, and Z. Q. Chen (2007), Magnetic field rotation analysis and the applications, J. Geophys. Res., 112, A06211, doi:10.1029/2005JA011584.

We have added the above comment and references to the revised manuscript.

2. "Later the Authors state: "Whereas the near-Earth magnetotail can become ballooning unstable under substorm conditions, the nonlinear evolution of ballooning instabilities, by themselves, may not lead to the near-explosive substorm onset." This statement is somewhat out of context. Do the Authors mean that coupling between ballooning and reconnection is necessary for explosive-like process?"

Reply: No, here we only meant to suggest that the coupling between ballooning and reconnection could be an alternative, though not necessary, route to the near-explosive substorm onset. We do not mean to completely rule out the possibility of explosive growth of nonlinear ballooning instability alone, however, its condition and in particular demonstration has remained a subject of research.

To avoid potential confusion, we have revised the statement as "Whereas the near-Earth magnetotail can become ballooning unstable under substorm conditions, the nonlinear evolution of ballooning instabilities, by themselves, may not always lead to the near-explosive growth. The coupling between ballooning and reconnection could be an alternative, though not the necessary, route to substorm onset." in the revised manuscript.

---

## Referee Comment (RC2) · Anonymous Referee #2 · 12 Mar 2019

In their paper the authors attempt to stress the importance of 3D geometry of the re-connection process through the QSL method that they adopt from the solar physics community and apply for MHD simulations of a magnetotail plasmoid formation in the presence of an MHD ballooning instability. The presented results are a very interesting peace of work, which is clear, novel and fit the scope of Ann. Geophys. This would be the reason for prompt publication. Although it would be nice to hear the authors' answers on two questions. 1) Would the QSL method be applicable for particle-in-cell simulations of reconnection caused in the course of a kinetic ballooning instability? 2) The authors showed that the reconnection sites identified with the QSL method were produced at the same x position and strictly periodically in the y direction. Thus, al-though the reconnection is three-dimentional, the resulting plasmoid, as far as I under-

stand is essentially a two-dimentional structure. This is hardly possible in nature and could be discussed a little in the last section of the paper.

---

## Author Comment (AC2) · 17 Mar 2019

We are grateful to the anonymous referee for the positive remarks and the suggestive questions on our work and manuscript. Below we address each of the questions:

Question 1) Would the QSL method be applicable for particle-in-cell simulations of reconnection caused in the course of a kinetic ballooning instability?

Reply: Yes, it should. The QSL is purely a geometric feature of the magnetic field configuration. Thus QSL method only relies on the magnetic field geometry in order to identify the reconnection sites. It is independent how the plasma is modelled, be it fluid or particle. Therefore the QSL method should be applicable for particle-in-cell simulations of reconnection caused in the course of a kinetic ballooning instability.

[Figure]

Question 2) The authors showed that the reconnection sites identified with the QSL method were produced at the same x position and strictly periodically in the y direction. Thus, although the reconnection is three-dimensional, the resulting plasmoid, as far as I understand is essentially a two-dimensional structure. This is hardly possible in nature and could be discussed a little in the last section of the paper.

Reply: Similar to the magnetic island, the plasmoid presented in this work is identified in the x-z plane as a finite region of closed magnetic flux bounded by a separatrix with a single X-point [e.g. Otto et al 1990, Zhu and Raeder 2014]. It is a two-dimensional projection onto the x-z plane of three-dimensional magnetic field lines in regions of magnetic reconnection. Whereas the plasmoid structure itself appears out of a two-dimensional projection, its occurrence in x-z plane is periodic in the y direction in our simulations, which indicates that the overall reconnecting field line structure is intrinsically 3D. Such a relation between the 2D plasmoid and the 3D reconnection is indeed possible, as demonstrated in our simulations, and may be more quantitatively captured in the 3D structure and distribution of QSLs.

We have added the above discussion to the last section of the revised manuscript.

A. Otto, K. Schindler, and J. Birn, Quantitative study of the nonlinear formation and acceleration of plasmoids in the Earth's magnetotail, J. Geophys. Res. 95, 15023-15037 (1990).

P. Zhu and J. Raeder, Ballooning instability-induced plasmoid formation in near-Earth plasma sheet, J. Geophys. Res. Space Physics 119, 131-141 (2014).